# Repeated Inverse Reinforcement Learning

**Kareem Amin**[*]
Google Research
New York, NY 10011
kamin@google.com

**Nan Jiang**[*]      **Satinder Singh**
Computer Science & Engineering,
University of Michigan, Ann Arbor, MI 48104
{nanjiang,baveja}@umich.edu

## Abstract

We introduce a novel *repeated* Inverse Reinforcement Learning problem: the agent has to act on behalf of a human in a sequence of tasks and wishes to minimize the number of tasks that it surprises the human by acting suboptimally with respect to how the human would have acted. Each time the human is surprised, the agent is provided a demonstration of the desired behavior by the human. We formalize this problem, including how the sequence of tasks is chosen, in a few different ways and provide some foundational results.

## 1   Introduction

One challenge in building AI agents that learn from experience is how to set their goals or rewards. In the Reinforcement Learning (RL) setting, one interesting answer to this question is inverse RL (or IRL) in which the agent infers the rewards of a human by observing the human's policy in a task [2]. Unfortunately, the IRL problem is ill-posed for there are typically many reward functions for which the observed behavior is optimal in a single task [3]. While the use of heuristics to select from among the set of feasible reward functions has led to successful applications of IRL to the problem of learning from demonstration [e.g., 4], not identifying the reward function poses fundamental challenges to the question of how well and how safely the agent will perform when using the learned reward function in other tasks.

We formalize multiple variations of a *new repeated IRL problem* in which the agent and (the same) human face multiple tasks over time. We separate the reward function into two components, one which is invariant across tasks and can be viewed as intrinsic to the human, and a second that is task specific. As a motivating example, consider a human doing tasks throughout a work day, e.g., getting coffee, driving to work, interacting with co-workers, and so on. Each of these tasks has a task-specific goal, but the human brings to each task intrinsic goals that correspond to maintaining health, financial well-being, not violating moral and legal principles, etc. In our repeated IRL setting, the agent presents a policy for each new task that it thinks the human would do. If the agent's policy "surprises" the human by being sub-optimal, the human presents the agent with the optimal policy. The objective of the agent is to minimize the number of surprises to the human, i.e., to generalize the human's behavior to new tasks.

In addition to addressing generalization across tasks, the repeated IRL problem we introduce and our results are of interest in resolving the question of unidentifiability of rewards from observations in standard IRL. Our results are also of interest to a particular aspect of the concern about how to make sure that the AI systems we build are safe, or AI safety. Specifically, the issue of reward misspecification is often mentioned in AI safety articles [e.g., 5, 6, 7]. These articles mostly discuss broad ethical concerns and possible research directions, while our paper develops mathematical formulations and algorithmic solutions to a specific way of addressing reward misspecification.

---

[*]This paper extends an unpublished arXiv paper by the authors [1].
[*]Equal contribution.

In summary form, our contributions include: (1) an efficient reward-identification algorithm when the agent can choose the tasks in which it observes human behavior; (2) an upper bound on the number of total surprises when no assumptions are made on the tasks, along with a corresponding lower bound; (3) an extension to the setting where the human provides sample trajectories instead of complete behavior; and (4) identification guarantees when the agent can only choose the task rewards but is given a fixed task environment.

## 2 Markov Decision Processes (MDPs)

An MDP is specified by its state space $\mathcal{S}$, action space $\mathcal{A}$, initial state distribution $\mu \in \Delta(\mathcal{S})$, transition function (or dynamics) $P : \mathcal{S} \times \mathcal{A} \to \Delta(\mathcal{S})$, reward function $Y : \mathcal{S} \to \mathbb{R}$, and discount factor $\gamma \in [0, 1)$. We assume finite $\mathcal{S}$ and $\mathcal{A}$, and $\Delta(\mathcal{S})$ is the space of all distributions over $\mathcal{S}$. A policy $\pi : \mathcal{S} \to \mathcal{A}$ describes an agent's behavior by specifying the action to take in each state. The (normalized) value function or long-term utility of $\pi$ is defined as $V^\pi(s) = (1-\gamma) \mathbb{E}[\sum_{t=1}^\infty \gamma^{t-1} Y(s_t) | s_0 = s; \pi].$[2] Similarly, the Q-value function is $Q^\pi(s, a) = (1 - \gamma) \mathbb{E}[\sum_{t=1}^\infty \gamma^{t-1} Y(s_t) | s_0 = s, a_0 = a; \pi]$. Where necessary we will use the notation $V_{P,Y}^\pi$ to avoid ambiguity about the dynamics and the reward function. Let $\pi^\star : \mathcal{S} \to \mathcal{A}$ be an optimal policy, which maximizes $V^\pi$ and $Q^\pi$ in all states (and actions) simultaneously.

Given an initial distribution over states, $\mu$, a scalar value that measures the goodness of $\pi$ is defined as $\mathbb{E}_{s \sim \mu}[V^\pi(s)]$. We introduce some further notation to express $\mathbb{E}_{s \sim \mu}[V^\pi(s)]$ in vector-matrix form. Let $\eta_{\mu,P}^\pi \in \mathbb{R}^{|\mathcal{S}|}$ be the *normalized state occupancy* under initial distribution $\mu$, dynamics $P$, and policy $\pi$, whose $s$-th entry is $(1-\gamma) \mathbb{E}[\sum_{t=1}^\infty \gamma^{t-1} \mathbb{I}(s_t = s) | s_0 \sim \mu; \pi]$ ($\mathbb{I}(\cdot)$ is the indicator function). This vector can be computed in closed-form as $\eta_{\mu,P}^\pi = (1 - \gamma) \left( \mu^\top P^\pi \left( \mathbf{I}_{|\mathcal{S}|} - \gamma P^\pi \right)^{-1} \right)^\top$, where $P^\pi$ is an $|\mathcal{S}| \times |\mathcal{S}|$ matrix whose $(s, s')$-th element is $P(s'|s, \pi(s))$, and $\mathbf{I}_{|\mathcal{S}|}$ is the $|\mathcal{S}| \times |\mathcal{S}|$ identity matrix. For convenience we will also treat the reward function $Y$ as a vector in $\mathbb{R}^{|\mathcal{S}|}$, and we have

$$\mathbb{E}_{s \sim \mu}[V^\pi(s)] = Y^\top \eta_{\mu,P}^\pi. \tag{1}$$

## 3 Problem setup

Here we define the **repeated IRL problem**. The human's reward function $\theta_\star$ captures his/her safety concerns and intrinsic/general preferences. This $\theta_\star$ is unknown to the agent and is the object of interest herein, i.e., if $\theta_\star$ were known to the agent, the concerns addressed in this paper would be solved. We assume that the human cannot directly communicate $\theta_\star$ to the agent but can evaluate the agent's behavior in a task as well as demonstrate optimal behavior. Each task comes with an external reward function $R$, and the goal is to maximize the reward with respect to $Y := \theta_\star + R$ in each task.

As a concrete example, consider an agent for an autonomous vehicle. In this case, $\theta_\star$ represents the cross-task principles that define good driving (e.g., courtesy towards pedestrians and other vehicles), which are often difficult to explicitly describe. In contrast, $R$, the task-specific reward, could reward the agent for successfully completing parallel parking. While $R$ is easier to construct, it may not completely capture what a human deems good driving. (For example, an agent might successfully parallel park while still boxing in neighboring vehicles.)

More formally, a task is defined by a pair $(E, R)$, where $E = (\mathcal{S}, \mathcal{A}, \mu, P, \gamma)$ is the task environment (i.e., a controlled Markov process) and $R$ is the task-specific reward function (*task reward*). We assume that all tasks share the same $\mathcal{S}, \mathcal{A}, \gamma$, with $|\mathcal{A}| \geq 2$, but **may** differ in the initial distribution $\mu$, dynamics $P$, and task reward $R$; all of the task-specifying quantities are known to the agent. In any task, the human's optimal behavior is always with respect to the reward function $Y = \theta_\star + R$. We emphasize again that $\theta_\star$ is intrinsic to the human and remains the same across all tasks. Our use of task specific reward functions $R$ allows for **greater generality** than the usual IRL setting, and most of our results apply equally to the case where $R \equiv \mathbf{0}$.

While $\theta_\star$ is private to the human, the agent has some prior knowledge on $\theta_\star$, represented as a set of possible parameters $\Theta_0 \subset \mathbb{R}^{|\mathcal{S}|}$ that contains $\theta_\star$. Throughout, we assume that the human's reward has bounded and normalized magnitude, that is, $\|\theta_\star\|_\infty \leq 1$.

A demonstration in $(E, R)$ reveals $\pi^\star$, optimal for $Y = \theta_\star + R$ under environment $E$, to the agent. A common assumption in the IRL literature is that the full mapping is revealed, which can be unrealistic if some states are unreachable from the initial distribution. We address the issue by requiring only the state occupancy vector $\eta_{\mu, P}^{\pi^*}$. In Section 7 we show that this also allows an easy extension to the setting where the human only demonstrates trajectories instead of providing a policy.

Under the above framework for repeated IRL, we consider two settings that differ in how the sequence of tasks are chosen. In both settings, we will want to minimize the number of demonstrations needed.

**1.** (Section 5) *Agent chooses the tasks*, observes the human's behavior in each of them, and infers the reward function. In this setting where the agent is powerful enough to choose tasks arbitrarily, we will show that the agent will be able to *identify* the human's reward function which of course implies the ability to generalize to new tasks.

**2.** (Section 6) *Nature chooses the tasks*, and the agent proposes a policy in each task. The human demonstrates a policy only if the agent's policy is significantly suboptimal (i.e., **a mistake**). In this setting we will derive upper and lower bounds on the number of mistakes our agent will make.

## 4 The challenge of identifying rewards

Note that it is impossible to identify $\theta_\star$ from watching human behavior in a single task. This is because any $\theta_\star$ is fundamentally indistinguishable from an infinite set of reward functions that yield exactly the policy observed in the task. We introduce the idea of *behavioral equivalence* below to tease apart two separate issues wrapped up in the challenge of identifying rewards.

**Definition 1.** Two reward functions $\theta, \theta' \in \mathbb{R}^{|\mathcal{S}|}$ are *behaviorally equivalent in all MDP tasks*, if for any $(E, R)$, the set of optimal policies for $(R + \theta)$ and $(R + \theta')$ are the same.

We argue that the task of identifying the reward function should amount only to identifying the (behavioral) equivalence class to which $\theta_\star$ belongs. In particular, identifying the equivalence class is sufficient to get perfect generalization to new tasks. Any remaining unidentifiability is merely representational and of no real consequence. Next we present a constraint that captures the reward functions that belong to the same equivalence class.

**Proposition 1.** *Two reward functions $\theta$ and $\theta'$ are behaviorally equivalent in all MDP tasks if and only if $\theta - \theta' = c \cdot \mathbf{1}_{|\mathcal{S}|}$ for some $c \in \mathbb{R}$, where $\mathbf{1}_{|\mathcal{S}|}$ is an all-1 vector of length $|\mathcal{S}|$.*

The proof is elementary and deferred to Appendix A. For any class of $\theta$'s that are equivalent to each other, we can choose a canonical element to represent this class. For example, we can fix an arbitrary reference state $s_{\text{ref}} \in \mathcal{S}$, and fix the reward of this state to 0 for $\theta_\star$ and all candidate $\theta$'s. In the rest of the paper, we will always assume such canonicalization in the MDP setting, hence $\theta_\star \in \Theta_0 \subseteq \{\theta \in [-1, 1]^{|\mathcal{S}|} : \theta(s_{\text{ref}}) = 0\}$.

## 5 Agent chooses the tasks

In this section, the protocol is that the agent chooses a sequence of tasks $\{(E_t, R_t)\}$. For each task $(E_t, R_t)$, the human reveals $\pi_t^\star$, which is optimal for environment $E_t$ and reward function $\theta_\star + R_t$. Our goal is to design an algorithm which chooses $\{(E_t, R_t)\}$ and identifies $\theta_\star$ to a desired accuracy, $\epsilon$, using as few tasks as possible. Theorem 1 shows that a simple algorithm can identify $\theta_\star$ after only $O(\log(1/\epsilon))$ tasks, if *any* tasks may be chosen. Roughly speaking, the algorithm amounts to a binary search on each component of $\theta_\star$ by manipulating the task reward $R_t$.[3] See the proof for the algorithm specification. As noted before, once the agent has identified $\theta_\star$ within an appropriate tolerance, it can compute a sufficiently-near-optimal policy for all tasks, thus completing the generalization objective through the far stronger identification objective in this setting.

**Theorem 1.** *If $\theta_\star \in \Theta_0 \subseteq \{\theta \in [-1, 1]^{|\mathcal{S}|} : \theta(s_{ref}) = 0\}$, there exists an algorithm that outputs $\theta \in \mathbb{R}^{|\mathcal{S}|}$ that satisfies $\|\theta - \theta_\star\|_\infty \leq \epsilon$ after $O(\log(1/\epsilon))$ demonstrations.*

*Proof.* The algorithm chooses the following fixed environment in all tasks: for each $s \in \mathcal{S} \setminus \{s_{\text{ref}}\}$, let one action be a self-loop, and the other action transitions to $s_{\text{ref}}$. In $s_{\text{ref}}$, all actions cause self-loops.

The initial distribution over states is uniformly at random over $\mathcal{S} \setminus \{s_{\text{ref}}\}$. Each task only differs in the task reward $R_t$ (where $R_t(s_{\text{ref}}) \equiv 0$ always). After observing the state occupancy of the optimal policy, for each $s$ we check if the occupancy is equal to 0. If so, it means that the demonstrated optimal policy chooses to go to $s_{\text{ref}}$ from $s$ in the first time step, and $\theta_\star(s) + R_t(s) \leq \theta_\star(s_{\text{ref}}) + R_t(s_{\text{ref}}) = 0$; if not, we have $\theta_\star(s) + R_t(s) \geq 0$. Consequently, after each task we learn the relationship between $\theta_\star(s)$ and $-R_t(s)$ on each $s \in \mathcal{S} \setminus \{s_{\text{ref}}\}$, so conducting a binary search by manipulating $R_t(s)$ will identify $\theta_\star$ to $\epsilon$-accuracy after $O(\log(1/\epsilon))$ tasks. □

## 6 Nature chooses the tasks

While Theorem 1 yields a strong identification guarantee, it also relies on a strong assumption, that $\{(E_t, R_t)\}$ may be chosen by the agent in an arbitrary manner. In this section, we let *nature*, who is allowed to be adversarial for the purpose of the analysis, choose $\{(E_t, R_t)\}$.

Generally speaking, we cannot obtain identification guarantees in such an adversarial setup. As an example, if $R_t \equiv 0$ and $E_t$ remains the same over time, we are essentially back to the classical IRL setting and suffer from the degeneracy issue. However, generalization to future tasks, which is our ultimate goal, is easy in this special case: after the initial demonstration, the agent can mimic it to behave optimally in all subsequent tasks without requiring further demonstrations. More generally, if nature repeats similar tasks, then the agent obtains little new information, but presumably it knows how to behave in most cases; if nature chooses a task unfamiliar to the agent, then the agent is likely to err, but it may learn about $\theta_\star$ from the mistake.

To formalize this intuition, we consider the following protocol: the nature chooses a sequence of tasks $\{(E_t, R_t)\}$ in an arbitrary manner. For every task $(E_t, R_t)$, the agent proposes a policy $\pi_t$. The human examines the policy's value under $\mu_t$, and if the loss

$$l_t = \mathbb{E}_{s \sim \mu}\left[V^{\pi_t^\star}_{E_t, \theta_\star + R_t}(s)\right] - \mathbb{E}_{s \sim \mu}\left[V^{\pi_t}_{E_t, \theta_\star + R_t}(s)\right] \tag{2}$$

is less than some $\epsilon$ then the human is satisfied and no demonstration is needed; otherwise a mistake is counted and $\eta^{\pi_t^\star}_{\mu_t, P_t}$ is revealed to the agent (note that $\eta^{\pi_t^\star}_{\mu_t, P_t}$ can be computed by the agent if needed from $\pi_t^*$ and its knowledge of the task). The main goal of this section is to design an algorithm that has a provable guarantee on the total number of mistakes.

**On human supervision** Here we require the human to evaluate the agent's policies in addition to providing demonstrations. We argue that this is a reasonable assumption because (1) only a binary signal $\mathbb{I}(l_t > \epsilon)$ is needed as opposed to the precise value of $l_t$, and (2) if a policy is suboptimal but the human fails to realize it, arguably it should not be treated as a mistake. Meanwhile, we will also provide identification guarantees in Section 6.4, as the human will be relieved from the supervision duty once $\theta_\star$ is identified.

Before describing and analyzing our algorithm, we first notice that the Equation 2 can be rewritten as

$$l_t = (\theta_\star + R)^\top (\eta^{\pi_t^\star}_{\mu_t, P_t} - \eta^{\pi_t}_{\mu_t, P_t}), \tag{3}$$

using Equation 1. So effectively, the given environment $E_t$ in each round induces a set of state occupancy vectors $\{\eta^\pi_{\mu_t, P_t} : \pi \in (\mathcal{S} \to \mathcal{A})\}$, and we want the agent to choose the vector that has the largest dot product with $\theta_\star + R$. The exponential size of the set will not be a concern because our main result (Theorem 2) has no dependence on the number of vectors, and only depends on the dimension of those vectors. The result is enabled by studying the *linear bandit* version of the problem, which subsumes the MDP setting for our purpose and is also a model of independent interest.

### 6.1 The linear bandit setting

In the linear bandit setting, $\mathcal{D}$ is a finite action space with size $|\mathcal{D}| = K$. Each task is denoted as a pair $(X, R)$, where $R$ is the task specific reward function as before. $X = [x^{(1)} \cdots x^{(K)}]$ is a $d \times K$ feature matrix, where $x^{(i)}$ is the feature vector for the $i$-th action, and $\|x^{(i)}\|_1 \leq 1$. When we reduce MDPs to linear bandits, each element of $\mathcal{D}$ corresponds to an MDP policy, and the feature vector is the state occupancy of that policy.

As before, $R, \theta_\star \in \mathbb{R}^d$ are the task reward and the human's unknown reward, respectively. The initial uncertainty set for $\theta_\star$ is $\Theta_0 \subseteq [-1, 1]^d$. The value of the $i$-th action is calculated as $(\theta_\star + R)^\top x^{(i)}$,

---

**Algorithm 1** Ellipsoid Algorithm for Repeated Inverse Reinforcement Learning

1: **Input:** $\Theta_0$.
2: $\Theta_1 \leftarrow \text{MVEE}(\Theta_0)$.
3: **for** $t = 1, 2, \ldots$ **do**
4:  Nature reveals $(X_t, R_t)$.
5:  Learner plays $a_t = \arg\max_{a \in \mathcal{D}} c_t^\top x_t^a$, where $c_t$ is the center of $\Theta_t$. $\Theta_{t+1} \leftarrow \Theta_t$.
6:  **if** $l_t > \epsilon$ **then**
7:   Human reveals $a_t^\star$. $\Theta_{t+1} \leftarrow \text{MVEE}(\{\theta \in \Theta_t : (\theta - c_t)^\top (x_t^{a_t^\star} - x_t^{a_t}) \geq 0\})$.
8:  **end if**
9: **end for**

---

and $a^\star$ is the action that maximizes this value. Every round the agent proposes an action $a \in \mathcal{D}$, whose loss is defined as

$$l_t = (\theta_\star + R)^\top (x^{a^\star} - x^a).$$

As before, a mistake is counted when $l_t > \epsilon$, in which case the optimal demonstration $x^{a^\star}$ is provided to the agent. We reiterate here that the agent only receives a binary signal $\mathbb{I}(l_t > \epsilon)$ in addition to the demonstration. We use the term "linear bandit" to refer to the generative process, but our interaction protocol differs from those in the standard bandit literature where reward or cost is revealed [8, 9].

We now show how to embed the previous MDP setting in the linear bandit setting.

**Example 1.** Given an MDP problem with variables $\mathcal{S}, \mathcal{A}, \gamma, \theta_\star, s_{\text{ref}}, \Theta_0, \{(E_t, R_t)\}$, we can convert it into a linear bandit problem as follows: (all variables with prime belong to the linear bandit problem, and we use $v^{\backslash i}$ to denote the vector $v$ with the $i$-th coordinate removed)

- $\mathcal{D} = \{\pi : \mathcal{S} \to \mathcal{A}\}, d = |\mathcal{S}| - 1, \theta_\star' = \theta_\star^{\backslash s_{\text{ref}}}, \Theta_0' = \{\theta^{\backslash s_{\text{ref}}} : \theta \in \Theta_0\}$.
- $x_t^\pi = (\eta_{\mu_t, P_t}^\pi)^{\backslash s_{\text{ref}}}. R_t' = R_t^{\backslash s_{\text{ref}}} - R_t(s_{\text{ref}}) \cdot \mathbf{1}_d$.

Note that there is a more straightforward conversion by letting $d = |\mathcal{S}|, \theta_\star' = \theta_\star, \Theta_0' = \Theta_0, x_t^\pi = \eta_{\mu_t, P_t}^\pi, R_t' = R_t$, which also preserves losses. We perform a more succinct conversion in Example 1 by canonicalizing both $\theta_\star$ (already assumed) and $R_t$ (explicitly done here) and dropping the coordinate for $s_{\text{ref}}$ in all relevant vectors.

**MDPs with linear rewards** In IRL literature, a generalization of the MDP setting is often considered, that reward is linear in state features $\phi(s) \in \mathbb{R}^d$ [2, 3]. In this new setting, $\theta_\star$ and $R$ are reward parameters, and the actual reward is $(\theta_\star + R)^\top \phi(s)$. This new setting can also be reduced to the linear bandit setting similarly to Example 1, except that the state occupancy is replaced by the discounted sum of expected feature values. Our main result, Theorem 2, will still apply automatically, but now the guarantee will only depend on the dimension of the feature space and has no dependence on $|\mathcal{S}|$. We include the conversion below but do not further discuss this setting in the rest of the paper.

**Example 2.** Consider an MDP problem with state features, defined by $\mathcal{S}, \mathcal{A}, \gamma, d \in \mathbb{Z}^+, \theta_\star \in \mathbb{R}^d, \Theta_0 \subseteq [-1, 1]^d, \{(E_t, \phi_t \in \mathbb{R}^d, R_t \in \mathbb{R}^d)\}$, where task reward and background reward in state $s$ are $\theta_\star^\top \phi_t(s)$ and $R^\top \phi_t(s)$ respectively, and $\theta_\star \in \Theta_0$. Suppose $\|\phi_t(s)\|_\infty \leq 1$ always holds, then we can convert it into a linear bandit problem as follows: $\mathcal{D} = \{\pi : \mathcal{S} \to \mathcal{A}\}$. $d, \theta_\star$, and $R_t$ remain the same. $x_t^\pi = (1 - \gamma) \sum_{h=1}^\infty \gamma^{h-1} \mathbb{E}[\phi(s_h) \mid \mu_t, P_t, \pi]/d$. Note that the division of $d$ in $x_t^\pi$ is for the purpose of normalization, so that $\|x_t^\pi\|_1 \leq \|\phi\|_1/d \leq \|\phi\|_\infty \leq 1$.

## 6.2 Ellipsoid Algorithm for Repeated Inverse Reinforcement Learning

We propose Algorithm 1, and provide the mistake bound in the following theorem.

**Theorem 2.** *For $\Theta_0 = [-1, 1]^d$, the number of mistakes made by Algorithm 1 is guaranteed to be $O(d^2 \log(d/\epsilon))$.*

To prove Theorem 2, we quote a result from linear programming literature in Lemma 1, which is found in standard lecture notes (e.g., [10], Theorem 8.8; see also [11], Lemma 3.1.34).

**Lemma 1** (Volume reduction in ellipsoid algorithm). *Given any non-degenerate ellipsoid $B$ in $\mathbb{R}^d$ centered at $c \in \mathbb{R}^d$, and any non-zero vector $v \in \mathbb{R}^d$, let $B^+$ be the minimum-volume enclosing ellipsoid (MVEE) of $\{u \in B : (u - c)^\top v \geq 0\}$. We have $vol(B^+)/vol(B) \leq e^{-\frac{1}{2(d+1)}}$.*

*Proof of Theorem 2.* Whenever a mistake is made, we can induce the constraint $(R_t + \theta_\star)^\top (x_t^{a_t^\star} - x_t^{a_t}) > \epsilon$. Meanwhile, since $a_t$ is greedy w.r.t. $c_t$, we have $(R_t + c_t)^\top (x_t^{a_t^\star} - x_t^{a_t}) \le 0$, where $c_t$ is the center of $\Theta_t$ as in Line 5. Taking the difference of the two inequalities, we obtain

$$(\theta_\star - c_t)^\top (x_t^{a_t^\star} - x_t^{a_t}) > \epsilon. \tag{4}$$

Therefore, the update rule on Line 7 of Algorithm 1 preserves $\theta_\star$ in $\Theta_{t+1}$. Since the update makes a central cut through the ellipsoid, Lemma 1 applies and the volume shrinks every time a mistake is made. To prove the theorem, it remains to upper bound the initial volume and lower bound the terminal volume of $\Theta_t$. We first show that an update never eliminates $B_\infty(\theta_\star, \epsilon/2)$, the $\ell_\infty$ ball centered at $\theta_\star$ with radius $\epsilon/2$. This is because, any eliminated $\theta$ satisfies $(\theta + c_t)^\top (x_t^{a_t^\star} - x_t^{a_t}) < 0$. Combining this with Equation 4, we have

$$\epsilon < (\theta^\star - \theta)^\top (x_t^{a_t^\star} - x_t^{a_t}) \le \|\theta_\star - \theta\|_\infty \|x_t^{a_t^\star} - x_t^{a_t}\|_1 \le 2\|\theta_\star - \theta\|_\infty.$$

The last step follows from $\|x\|_1 \le 1$. We conclude that any eliminated $\theta$ should be $\epsilon/2$ far away from $\theta_\star$ in $\ell_\infty$ distance. Hence, we can lower bound the volume of $\Theta_t$ for any $t$ by that of $\Theta_0 \bigcap B_\infty(\theta_\star, \epsilon/2)$, which contains an $\ell_\infty$ ball with radius $\epsilon/4$ at its smallest (when $\theta_\star$ is one of $\Theta_0$'s vertices). To simplify calculation, we relax this lower bound (volume of the $\ell_\infty$ ball) to the volume of the inscribed $\ell_2$ ball.

Finally we put everything together: let $M_T$ be the number of mistakes made from round 1 to $T$, $C_d$ be the volume of the unit hypersphere in $\mathbb{R}^d$ (i.e., $\ell_2$ ball with radius 1), and $\text{vol}(\cdot)$ denote the volume of an ellipsoid, we have

$$\frac{M_T}{2(d+1)} \le \log(\text{vol}(\Theta_1)) - \log(\text{vol}(\Theta_{T+1})) \le \log(C_d(\sqrt{d})^d) - \log(C_d(\epsilon/4)^d) = d \log \frac{4\sqrt{d}}{\epsilon}.$$

So $M_T \le 2d(d+1) \log \frac{4\sqrt{d}}{\epsilon} = O(d^2 \log \frac{d}{\epsilon})$. $\qquad\square$

## 6.3 Lower bound

In Section 5, we get an $O(\log(1/\epsilon))$ upper bound on the number of demonstrations, which has no dependence on $|\mathcal{S}|$ (which corresponds to $d+1$ in the linear bandit setting). Comparing Theorem 2 to 1, one may wonder whether the polynomial dependence on $d$ is an artifact of the inefficiency of Algorithm 1. We clarify this issue by proving a lower bound, showing that $\Omega(d \log(1/\epsilon))$ mistakes are inevitable in the worst case when nature chooses the tasks. We provide a proof sketch below, and the complete proof is deferred to Appendix E.

**Theorem 3.** *For any randomized algorithm[4] in the linear bandit setting, there always exists $\theta_\star \in [-1, 1]^d$ and an adversarial sequence of $\{(X_t, R_t)\}$ that potentially adapts to the algorithm's previous decisions, such that the expected number of mistakes made by the algorithm is $\Omega(d \log(1/\epsilon))$.*

*Proof Sketch.* We randomize $\theta_\star$ by sampling each element i.i.d. from $\text{Unif}([-1, 1])$. We will prove that there exists a strategy of choosing $(X_t, R_t)$ such that any algorithm's expected number of mistakes is $\Omega(d \log(1/\epsilon))$, which proves the theorem as max is no less than average.

In our construction, $X_t = [\mathbf{0}_d, \ e_{j_t}]$, where $j_t$ is some index to be specified. Hence, every round the agent is essentially asked to decided whether $\theta(j_t) \ge -R_t(j_t)$. The adversary's strategy goes in phases, and $R_t$ remains the same during each phase. Every phase has $d$ rounds where $j_t$ is enumerated over $\{1, \dots, d\}$.

The adversary will use $R_t$ to shift the posterior on $\theta(j_t) + R_t(j_t)$ so that it is centered around the origin; in this way, the agent has about $1/2$ probability to make an error (regardless of the algorithm), and the posterior interval will be halved. Overall, the agent makes $d/2$ mistakes in each phase, and there will be about $\log(1/\epsilon)$ phases in total, which gives the lower bound. $\qquad\square$

**Applying the lower bound to MDPs** The above lower bound is stated for the linear bandit setting. In principle, we need to prove lower bound for MDPs separately, because linear bandits are more general than MDPs for our purpose, and the hard instances in linear bandits may not have corresponding

MDP instances. In Lemma 2 below, we show that a certain type of linear bandit instances can always be emulated by MDPs with the same number of actions, and the hard instances constructed in Theorem 3 indeed satisfy the conditions for such a type; in particular, we require the feature vectors to be non-negative and have $\ell_1$ norm bounded by 1. As a corollary, an $\Omega(|\mathcal{S}| \log(1/\epsilon))$ lower bound for the MDP setting (even with a small action space $|\mathcal{A}| = 2$) follows directly from Theorem 3. The proof of Lemma 2 is deferred to Appendix B.

**Lemma 2** (Linear bandit to MDP conversion). *Let $(X, R)$ be a linear bandit task, and $K$ be the number of actions. If every $x^a$ is non-negative and $\|x^a\|_1 \leq 1$, then there exists an MDP task $(E, R')$ with $d + 1$ states and $K$ actions, such that under some choice of $s_{ref}$, converting $(E, R')$ as in Example 1 recovers the original problem.*

### 6.4 On identification when nature chooses tasks

While Theorem 2 successfully controls the number of total mistakes, it completely avoids the identification problem and does not guarantee to recover $\theta_\star$. In this section we explore further conditions under which we can obtain identification guarantees when Nature chooses the tasks.

The first condition, stated in Proposition 2, implies that if we have made all the possible mistakes, then we have indeed identified the $\theta_\star$, where the identification accuracy is determined by the tolerance parameter $\epsilon$ that defines what is counted as a mistake. Due to space limit, the proof is deferred to Appendix C.

**Proposition 2.** *Consider the linear bandit setting. If there exists $T_0$ such that for any round $t \geq T_0$, no more mistakes can be ever made by the algorithm for any choice of $(E_t, R_t)$ and any tie-braking mechanism, then we have $\theta_\star \in B_\infty(c_{T_0}, \epsilon)$.*

While the above proposition shows that identification is guaranteed if the agent exhausts the mistakes, the agent has no ability to actively fulfill this condition when nature chooses tasks. For a stronger identification guarantee, we may need to grant the agent some freedom in choosing the tasks.

**Identification with fixed environment**  Here we consider a setting that fits in between Section 5 (completely active) and Section 6.1 (completely passive), where the environment $E$ (hence the induced feature vectors $\{x^{(1)}, x^{(2)}, \ldots, x^{(K)}\}$) is given and fixed, and the agent can arbitrarily choose the task reward $R_t$. The goal is to obtain identification guarantee in this intermediate setting.

Unfortunately, a degenerate case can be easily constructed that prevents the revelation of any information about $\theta_\star$. In particular, if $x^{(1)} = x^{(2)} = \ldots = x^{(K)}$, i.e., the environment is completely uncontrolled, then all actions are equally optimal and nothing can be learned. More generally, if for some $v \neq \mathbf{0}$ we have $v^\top x^{(1)} = v^\top x^{(2)} = \ldots = v^\top x^{(K)}$, then we may never recover $\theta_\star$ along the direction of $v$. In fact, Proposition 1 can be viewed as an instance of this result where $v = \mathbf{1}_{|\mathcal{S}|}$ (recall that $\mathbf{1}_{|\mathcal{S}|}^\top \eta_{\mu,P}^\pi \equiv 1$), and that is why we have to remove such redundancy in Example 1 in order to discuss identification in MDPs. Therefore, to guarantee identification in a fixed environment, the feature vectors must have significant variation in all directions, and we capture this intuition by defining a diversity score spread$(X)$ (Definition 2) and showing that the identification accuracy depends inversely on the score (Theorem 4).

**Definition 2.** Given the feature matrix $X = \begin{bmatrix} x^{(1)} & x^{(2)} & \cdots & x^{(K)} \end{bmatrix}$ whose size is $d \times K$, define spread$(X)$ as the $d$-th largest singular value of $\widetilde{X} := X(\mathbf{I}_K - \frac{1}{K}\mathbf{1}_K\mathbf{1}_K^\top)$.

**Theorem 4.** *For a fixed feature matrix $X$, if spread$(X) > 0$, then there exists a sequence $R_1, R_2, \ldots, R_T$ with $T = O(d^2 \log(d/\epsilon))$ and a sequence of tie-break choices of the algorithm, such that after round $T$ we have $\|c_T - \theta_\star\|_\infty \leq \epsilon\sqrt{(K-1)/2}/spread(X)$.*

The proof is deferred to Appendix D. The $\sqrt{K}$ dependence in Theorem 4 may be of concern as $K$ can be exponentially large. However, Theorem 4 also holds if we replace $X$ by any matrix that consists of $X$'s columns, so we may choose a small yet most diverse set of columns as to optimize the bound.

## 7 Working with trajectories

In previous sections, we have assumed that the human evaluates the agent's performance based on the state occupancy of the agent's policy, and demonstrates the optimal policy in terms of state occupancy

---
**Algorithm 2** Trajectory version of Algorithm 1 for MDPs
---
1: **Input:** $\Theta_0, H, n$.
2: $\Theta_1 \leftarrow \text{MVEE}(\Theta_0)$, $i \leftarrow 0$, $\bar{Z} \leftarrow 0$, $\bar{Z}^\star \leftarrow 0$.
3: **for** $t = 1, 2, \ldots$ **do**
4:      Nature reveals $(E_t, R_t)$. Agent rolls-out a trajectory using $\pi_t$ greedily w.r.t. $c_t + R_t$.
5:      $\Theta_{t+1} \leftarrow \Theta_t$.
6:      **if** agent takes $a$ in $s$ with $Q^\star(s, a) < V^\star(s) - \epsilon$ **then**
7:          Human produces an $H$-step trajectory from $s$. Let the empirical state occupancy be $\hat{z}_i^{\star, H}$.
8:          $i \leftarrow i + 1$, $\bar{Z}^\star \leftarrow \bar{Z}^\star + \hat{z}_i^{\star, H}$.
9:          Let $z_i$ be the state occupancy of $\pi_t$ from initial state $s$, and $\bar{Z} \leftarrow \bar{Z} + z_i$.
10:         **if** $i = n$ **then**
11:            $\Theta_{t+1} \leftarrow \text{MVEE}(\{\theta \in \Theta_t : (\theta - c_t)^\top (\bar{Z}^\star - \bar{Z}) \geq 0\})$.    $i \leftarrow 0$, $\bar{Z} \leftarrow 0$, $\bar{Z}^\star \leftarrow 0$.
12:         **end if**
13:      **end if**
14: **end for**
---

as well. In practice, we would like to instead assume that for each task, the agent rolls out a trajectory, and the human shows an optimal trajectory if he/she finds the agent's trajectory unsatisfying. We are still concerned about upper bounding the number of total mistakes, and aim to provide a parallel version of Theorem 2.

Unlike in traditional IRL, in our setting the agent is also acting, which gives rise to many subtleties. First, the total reward on the agent's single trajectory is a random variable, and may deviate from the expected value of its policy. Therefore, it is generally impossible to decide if the agent's policy is near-optimal, and instead we assume that the human can check if each action that the agent takes in the trajectory is near-optimal: when the agent takes $a$ at state $s$, an error is counted if and only if $Q^\star(s, a) < V^\star(s) - \epsilon$. This criterion can be viewed as a noisy version of the one used in previous sections, as taking expectation of $V^\star(s) - Q^\star(s, \pi(s))$ over the occupancy induced by $\pi$ will recover Equation 2.

While this resolves the issue on the agent's side, how should the human provide his/her optimal trajectory? The most straightforward protocol is that the human rolls out a trajectory from the initial distribution of the task, $\mu_t$. We argue that this is not a reasonable protocol for two reasons: (1) in expectation, the reward collected by the human may be less than that by the agent, because conditioning on the event that an error is spotted may introduce a selection bias; (2) the human may not encounter the problematic state in his/her own trajectory, hence the information provided in the trajectory may be irrelevant.

To resolve this issue, we consider a different protocol where the human rolls out a trajectory using an optimal policy from the very state where the agent errs.

Now we discuss how we can prove a parallel of Theorem 2 under this new protocol. First, let's assume that the demonstration were still given in the form a state occupancy vector starting at the problematic state. In this case, we can reduce to the setting of Section 6 by changing $\mu_t$ to a point mass on the problematic state.[5] To apply the algorithm and the analysis in Section 6, it remains to show that the notion of error in this section (a suboptimal action) implies the notion of error in Section 6 (a suboptimal policy): let $s$ be the problematic state and $\pi$ be the agent's policy, we have $V^\pi(s) = Q^\pi(s, \pi(s)) \leq Q^\star(s, \pi(s)) < V^\star(s) - \epsilon$. So whenever a suboptimal *action* is spotted in state $s$, it indeed implies that the agent's *policy* is suboptimal for $s$ as the initial state. Hence, we can run Algorithm 1 as-is and Theorem 2 immediately applies.

To tackle the remaining issue that the demonstration is in terms of a single trajectory, we will not update $\Theta_t$ after each mistake as in Algorithm 1, but only make an update after every mini-batch of mistakes, and aggregate them to form accurate update rules. See Algorithm 2. The formal guarantee of the algorithm is stated in Theorem 5, whose proof is deferred to Appendix G.

**Theorem 5.** $\forall \delta \in (0, 1)$*, with probability at least* $1 - \delta$*, the number of mistakes made by Algorithm 2 with parameters* $\Theta_0 = [-1, 1]^d$*,* $H = \left\lceil \frac{\log(12/\epsilon)}{1-\gamma} \right\rceil$*, and* $n = \left\lceil \frac{\log(\frac{4d(d+1) \log \frac{6\sqrt{d}}{\epsilon}}{\delta})}{32\epsilon^2} \right\rceil$ *where* $d = |\mathcal{S}|$,[6] *is at most* $\tilde{O}(\frac{d^2}{\epsilon^2} \log(\frac{d}{\delta \epsilon}))$.[7]

## 8 Related work & Conclusions

Most existing work in IRL focused on inferring the reward function[8] using data acquired from a fixed environment [2, 3, 18, 19, 20, 21, 22]. There is prior work on using data collected from multiple — but exogenously fixed — environments to predict agent behavior [23]. There are also applications where methods for single-environment MDPs have been adapted to multiple environments [19]. Nevertheless, all these works consider the objective of mimicking an optimal behavior in the presented environment(s), and do not aim at generalization to new tasks that is the main contribution of this paper. Recently, Hadfield-Menell et al. [24] proposed cooperative inverse reinforcement learning, where the human and the agent act in the same environment, allowing the human to actively resolve the agent's uncertainty on the reward function. However, they only consider a single environment (or task), and the unidentifiability issue of IRL still exists. Combining their interesting framework with our resolution to unidentifiability (by multiple tasks) can be an interesting future direction.

## Acknowledgement

This work was supported in part by NSF grant IIS 1319365 (Singh & Jiang) and in part by a Rackham Predoctoral Fellowship from the University of Michigan (Jiang). Any opinions, findings, conclusions, or recommendations expressed here are those of the authors and do not necessarily reflect the views of the sponsors.

## Footnotes

[2]Here we differ (w.l.o.g.) from common IRL literature in assuming that reward occurs after transition.

[3]While we present a proof that manipulates $R_t$, an only slightly more complex proof applies to the setting where all the $R_t$ are exactly zero and the manipulation is limited to the environment [1].

[4]While our Algorithm 1 is deterministic, randomization is often crucial for online learning in general [12].

[5]At the first glance this might seem suspicious: the problematic state is random and depends on the learner's current policy, but in RL the initial distribution is usually fixed and the learner has no control over it. This concern is removed thanks to our adversarial setup on $(E_t, R_t)$ (of which $\mu_t$ is a component).

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
