[Supplementary Material]

# Appendix

## A  Proof of Proposition 1

To show that $\theta - \theta' = c \cdot \mathbf{1}_{|\mathcal{S}|}$ implies behavioral equivalence, we note that for any policy $\pi$ the occupancy vector $\eta_{\mu,P}^{\pi}$ always satisfies $\mathbf{1}_{|\mathcal{S}|}^{\top} \eta_{\mu,P}^{\pi} = 1$, so $\forall \pi, |\theta^T \eta_{\mu,P}^{\pi} - \theta'^T \eta_{\mu,P}^{\pi}| = c$, and therefore the set of optimal policies is the same.

To show the other direction, we prove that if $\theta - \theta' \notin \mathrm{span}(\{\mathbf{1}_{|\mathcal{S}|}\})$, then there exists $(E, R)$ such that the sets of optimal policies differ. In particular, we choose $R = -\theta'$, so that all policies are optimal under $R + \theta' = 0$. Since $\theta - \theta' \notin \mathrm{span}(\{\mathbf{1}_{|\mathcal{S}|}\})$, there exists states $i$ and $j$ such that $\theta(i) + R(i) \neq \theta(j) + R(j)$. Suppose $i$ is the one with smaller sum of rewards, then we can make $j$ an absorbing state, and have two deterministic actions in $i$ that transition to $i$ and $j$ respectively. Under $R + \theta$, the self-loop in state $i$ is suboptimal, and this completes the proof. $\qquad \square$

## B  Proof of Lemma 2

The construction is as follows. Choose $s_{\mathrm{ref}}$ as the initial state, and make all other states absorbing. Let $R'(s_{\mathrm{ref}}) = 0$ and $R'$ restricted on $\mathcal{S} \setminus \{s_{\mathrm{ref}}\}$ coincide with $R$. The remaining work is to design the transition distribution of each action in $s_{\mathrm{ref}}$ so that the induced state occupancy matches exactly one column of $X$.

Fixing any action $a$, and let $x$ be the feature that we want to associate $a$ with. The next-state distribution of $(s_{\mathrm{ref}}, a)$ is as follows: with probability $p = \frac{1 - \|x\|_1}{1 - \gamma\|x\|_1}$ the next-state is $s_{\mathrm{ref}}$ itself, and the probability of transitioning to the $j$-th state in $\mathcal{S} \setminus \{s_{\mathrm{ref}}\}$ is $\frac{1-\gamma}{1-\gamma\|x\|_1} x(j)$. Given $\|x\|_1 \leq 1$ and $x \geq 0$, it is easy to verify that this is a valid distribution.

Now we calculate the occupancy of policy $\pi(s_{\mathrm{ref}}) = a$. The normalized occupancy on $s_{\mathrm{ref}}$ is

$$(1-\gamma)(p + \gamma p^2 + \gamma^2 p^3 + \cdots) = \frac{p(1-\gamma)}{1 - \gamma p} = 1 - \|x\|_1.$$

The remaining occupancy, with a total $\ell_1$ mass of $\|x\|_1$, is split among $\mathcal{S} \setminus \{s_{\mathrm{ref}}\}$ proportional to $x$. Therefore, when we convert the MDP problem as in Example 1, the corresponding feature vector is exactly $x$, so we recover the original linear bandit problem. $\qquad \square$

## C  Proof of Proposition 2

Assume towards contradiction that $\|c_{T_0} - \theta_\star\|_\infty > \epsilon$. We will choose $(R_t, x_t^{(1)}, x_t^{(2)})$ to make the algorithm err. In particular, let $R_t = -c_{T_0}$, so that the algorithm acts greedily with respect to $\mathbf{0}_d$. Since $\mathbf{0}_d^\top x_t^a \equiv 0$, any action would be a valid choice for the algorithm.

On the other hand, $\|c_{T_0} - \theta_\star\|_\infty > \epsilon$ implies that there exists a coordinate $j$ such that $|e_j^\top(\theta_\star - c_{T_0})| > \epsilon$, where $e_j$ is a basis vector. Let $x_t^{(1)} = \mathbf{0}_d$ and $x_t^{(2)} = e_j$. So the value of action 1 is always 0 under any reward function (including $\theta_\star + R_t$), and the value of action 2 is $(\theta_\star + R_t)^\top x_t^{(2)} = (\theta_\star - c_{T_0})^\top e_j$, whose absolute value is greater than $\epsilon$. At least one of the 2 actions is more than $\epsilon$ suboptimal, and the algorithm may take any of them, so the algorithm can err again. $\qquad \square$

## D  Proof of Theorem 4

It suffices to show that in any round $t$, if $\|c_t - \theta_\star\|_\infty > \frac{\epsilon\sqrt{(K-1)/2}}{\mathrm{spread}(X)}$, then $l_t > \epsilon$. The bound on $T$ follows directly from Theorem 2. Similar to the proof of Proposition 2, our choice of the task reward is $R_t = -c_t$, so that any $a \in A$ would be a valid choice of $a_t$, and we will choose the worst action. Note that $\forall a, a' \in \mathcal{D}$,

$$l_t = (\theta_\star + R_t)^\top (x^{a_t^\star} - x^{a_t}) \geq (\theta_\star - c_t)^\top (x^a - x^{a'}).$$

So it suffices to show that there exists $a, a' \in \mathcal{D}$, such that $(\theta_\star - c_t)^\top (x^a - x^{a'}) > \epsilon$. Let $y_t = \theta_\star - c_t$, and the precondition implies that $\|y_t\|_2 \geq \|y_t\|_\infty > \frac{\epsilon\sqrt{(K-1)/2}}{\mathrm{spread}(X)}$.

Define a matrix $D$ of size $K \times (K(K-1))$, where each column

$$D = \begin{bmatrix} 1 & 1 & \cdots & 0 \\ -1 & 0 & \cdots & 0 \\ 0 & -1 & \cdots & 0 \\ & & \ddots & \\ 0 & 0 & \cdots & -1 \\ 0 & 0 & \cdots & 1 \end{bmatrix}. \tag{5}$$

contains exactly one 1 and one $-1$ (the remaining entries are 0), and the columns enumerate all possible positions of them. With the help of this matrix, we can rewrite the desired result $(\exists\, a, a' \in A, \text{ s.t. } (\theta_\star - c_t)^\top (x^a - x^{a'}) > \epsilon)$ as $\|y_t^\top X D\|_\infty \geq \epsilon$. We relax the LHS as $\|y_t^\top X D\|_\infty \geq \|y_t^\top X D\|_2 / \sqrt{K(K-1)}$, and will provide a lower bound on $\|y_t^\top X D\|_2$. Note that

$$y_t^\top X D = y_t^\top (\widetilde{X} + (X - \widetilde{X}))D = y_t^\top \widetilde{X} D,$$

because every row of $(X - \widetilde{X})$ is some multiple of $\mathbf{1}_K^\top$ (recall Definition 2), and every column of $D$ is orthogonal to $\mathbf{1}_K$. Let $\widehat{(\cdot)}$ be the vector normalized to unit length,

$$\|y_t^\top \widetilde{X} D\|_2 = \|y_t\|_2 \|\widehat{y}_t^\top \widetilde{X} D\|_2 = \|y_t\|_2 \|\widehat{y}_t^\top \widetilde{X}\|_2 \|\widehat{\widehat{y}_t^\top \widetilde{X}} D\|_2.$$

We lower bound each of the 3 terms. For the first term, we have the precondition $\|y_t\|_2 > \frac{\epsilon\sqrt{(K-1)/2}}{\mathrm{spread}(X)}$. The second term is $\widetilde{X}$ left multiplied by a unit vector, so its $\ell_2$ norm can be lower bounded by the smallest non-zero singular value of $\widetilde{X}$ (recall that $\widetilde{X}$ is full-rank), which is $\mathrm{spread}(X)$.

To lower bound the last term, note that $DD^\top = 2K\mathbf{I}_K - 2\mathbf{1}_K\mathbf{1}_K^\top$, and rows of $\widetilde{X}$ are orthogonal to $\mathbf{1}_K^\top$ and so is $y_t^\top \widetilde{X}$, so

$$\|\widehat{\widehat{y}_t^\top \widetilde{X}} D\|_2^2 \geq \inf_{\|z\|_2=1,\, z \perp \mathbf{1}_K} z^\top DD^\top z = \inf_{\|z\|_2=1,\, z \perp \mathbf{1}_K} z^\top (2K\mathbf{I}_K - 2\mathbf{1}_K\mathbf{1}_K^\top)z = 2K.$$

Putting all the pieces together, we have

$$\|y_t^\top \widetilde{X} D\|_\infty \geq \frac{\|y_t\|_2 \|\widehat{y}_t^\top \widetilde{X}\|_2 \|\widehat{\widehat{y}_t^\top \widetilde{X}} D\|_2}{\sqrt{K(K-1)}} > \frac{\epsilon\sqrt{(K-1)/2}}{\mathrm{spread}(X)} \cdot \mathrm{spread}(X) \cdot \frac{\sqrt{2K}}{\sqrt{K(K-1)}} = \epsilon.$$

## E  Proof of Theorem 3

As a standard trick, we randomize $\theta_\star$ by sampling each element i.i.d. from $\mathrm{Unif}([-1, 1])$. We will prove that there exists a strategy of choosing $(X_t, R_t)$ such that any algorithm's expected number of mistakes is $\Omega(d\log(1/\epsilon))$, where the expectation is with respect to the randomness of $\theta_\star$ and the internal randomness of the algorithm. This immediately implies a worst-case result as max is no less than average (regarding the sampling of $\theta_\star$).

In our construction, $X_t = [\mathbf{0}_d,\, e_{j_t}]$, where $j_t$ is some index to be specified. Hence, every round the agent is essentially asked to decided whether $\theta(j_t) \geq -R_t(j_t)$. The adversary's strategy goes in phases, and $R_t$ remains the same during each phase. Every phase has $d$ rounds where $j_t$ is enumerated over $\{1, \ldots, d\}$. To fully specify the nature's strategy, it remains to specify $R_t$ for each phase.

In the 1st phase, $R_t \equiv 0$. For each coordinate $j$, the information revealed to the agent is one of the following: $\theta_\star(j) > \epsilon$, $\theta_\star(j) \geq -\epsilon$, $\theta_\star(j) < -\epsilon$, $\theta_\star(j) \leq \epsilon$. For clarity we first make an simplification, that the revealed information is either $\theta_\star(j) > 0$ or $\theta_\star(j) \leq 0$; we will deal with the subtleties related to $\epsilon$ at the end of the proof.

In the 2nd phase, we fix $R_t$ as

$$R_t(j) = \begin{cases} -1/2 & \text{if } \theta_\star(j) \geq 0, \\ 1/2 & \text{if } \theta_\star(j) < 0. \end{cases}$$

Since $\theta_\star$ is randomized i.i.d. for each coordinate, the posterior of $\theta_\star + R_t$ conditioned on the revealed information is $\text{Unif}[-1/2, 1/2]$, for any algorithm and any interaction history. Therefore the 2nd phase is almost identical to the 1st phase except that the intervals have shrunk by a factor of 2. Similarly in the 3rd phase we use $R_t$ to offset the posterior of $\theta_\star + R_t$ to $\text{Unif}([-1/4, 1/4])$, and so on.

In phase $m$, the half-length of the interval is $2^{-m+1}$, and the probability that a mistake occurs is at least $1/2 - \epsilon/2^{-m+2}$ for any algorithm. The whole process continues as long as this probability is greater than 0. By linearity of expectation, we can lower bound the total mistakes by the sum of expected mistakes in each phase, which gives

$$\sum_{2^{-m+1} \geq \epsilon} d(1/2 - \epsilon/2^{-m+2}) \geq \sum_{2^{-m+1} \geq 2\epsilon} d \cdot 1/4 \geq \lfloor \log_2(1/\epsilon) \rfloor d/4. \tag{6}$$

The above analysis made a simplification that the posterior of $\theta_\star + R_t$ in phase $m$ is $[-2^{-m+1}, 2^{-m+1}]$. We now remove the simplification. Note, however, that if we choose $R_t$ to center the posterior, $R_t$ reveals no additional information about $\theta_\star$, and in the worst case the interval shrinks to half of its previous size minus $\epsilon$. So the length of interval in phase $m$ is at least $2^{-m+2}(1 + \epsilon) - 2\epsilon$, and the error probability is at least $1/2 - \epsilon/(2^{-m+1}(1 + \epsilon) - \epsilon)$. The rest of the analysis is similar: we count the number of phases until the error probability drops below $1/4$, and in each of these phases we get at least $d/4$ mistakes in expectation. The number of such phases is given by

$$1/2 - \epsilon/(2^{-m+1}(1 + \epsilon) - \epsilon) \geq 1/4,$$

which is satisfied when $2^{-m+1} \geq 5\epsilon$, that is, when $m \leq \lfloor \log_2 \frac{2}{5\epsilon} \rfloor$. This completes the proof. $\square$

## F   Bounding the $\ell_\infty$ distance between $\theta_\star$ and the ellipsoid center

To prove Theorem 5, we need an upper bound on $\|\theta_\star - c\|_\infty$ for quantifying the error due to $H$-step truncation and sampling effects, where $c$ is the ellipsoid center. As far as we know there is no standard result on this issue. However, a simple workaround, described below, allows us to assume $\|\theta_\star - c\|_\infty \leq 2$ without loss of generality.

Whenever $\|c\|_\infty > 1$, there exists coordinate $j$ such that $|c_j| > 1$. We can make a central cut $e_j^\top(\theta - c) < 0$ (or $> 0$ depending on the sign of $c_j$), and replace the original ellipsoid with the MVEE of the remaining shape. This operation never excludes any point in $\Theta_0$, hence it allows the proofs of Theorem 2 and 5 to work. We keep making such cuts and update the ellipsoid accordingly, until the new center satisfies $\|c\|_\infty \leq 1$. Since central cuts reduce volume substantially (Lemma 1) and there is a lower bound on the volume, the process must stop after finite number of operations. After the process stops, we have $\|\theta_\star - c\|_\infty \leq \|\theta_\star\|_\infty + \|c\|_\infty \leq 2$.

## G   Proof of Theorem 5

We first introduce a standard concentration inequality for martingales.

**Lemma 3** (Azuma's inequality for martingales). *Suppose $\{S_0, S_1, \ldots, S_n\}$ is a martingale and $|S_i - S_{i-1}| \leq b$ almost surely. Then with probability at least $1 - \delta$ we have $|S_n - S_0| \leq b\sqrt{2n \log(2/\delta)}$.*

*Proof.* Since the update rule is still in the format of a central cut through the ellipsoid, Lemma 1 applies. It remains to show that the update rule preserves $\theta_\star$ and a certain volume around it, and then we can follow the same argument as for Theorem 2.

Fixing a mini-batch, let $t_0$ be the round on which the last update occurs, and $\Theta = \Theta_{t_0}, c = c_{t_0}$. Note that $\Theta_t = \Theta$ during the collection of the current mini-batch and does not change, and $c_t = c$ similarly.

For each $i = 1, 2, \ldots, n$, define $z_i^{\star, H}$ as the expected value of $\hat{z}_i^{\star, H}$, where expectation is with respect to the randomness of the trajectory produced by the human, and let $z_i^\star$ be the infinite-step expected state occupancy. Note that $\hat{z}_i^{\star, H}, z_i^{\star, H}, z_i^\star \in \mathbb{R}^{|\mathcal{S}|}$.

As before, we have $\theta_\star^\top(z_i^\star - z_i) > \epsilon$ and $c^\top(z_i^\star - z_i) \leq 0$, so $(\theta_\star - c)^\top(z_i^\star - z_i) > \epsilon$. Taking average over $i$, we get $(\theta_\star - c)^\top(\frac{1}{n}\sum_{i=1}^n z_i^\star - \frac{1}{n}\sum_{i=1}^n z_i) > \epsilon$.

What we will show next is that $(\theta_\star - c)^\top(\frac{\bar{Z}^\star}{n} - \frac{\bar{Z}}{n}) > \epsilon/3$ for $\bar{Z}^\star$ and $\bar{Z}$ on Line 11, which implies that the update rule is valid and has enough slackness for lower bounding the volume of $\Theta_t$ as before. Note that

$$
\begin{aligned}
(\theta_\star - c)^\top(\tfrac{\bar{Z}^\star}{n} - \tfrac{\bar{Z}}{n}) &= (\theta_\star - c)^\top(\tfrac{1}{n}\textstyle\sum_{i=1}^n z_i^\star - \tfrac{1}{n}\textstyle\sum_{i=1}^n z_i) \\
&- (\theta_\star - c)^\top(\tfrac{1}{n}\textstyle\sum_{i=1}^n z_i^\star - \tfrac{1}{n}\textstyle\sum_{i=1}^n z_i^{\star,H}) \\
&- (\theta_\star - c)^\top(\tfrac{1}{n}\textstyle\sum_{i=1}^n z_i^{\star,H} - \tfrac{1}{n}\textstyle\sum_{i=1}^n \hat{z}_i^{\star,H}).
\end{aligned}
$$

Here we decompose the expression of interest into 3 terms. The 1st term is lower bounded by $\epsilon$ as shown above, and we will upper bound each of the remaining 2 terms by $\epsilon/3$. For the 2nd term, since $\|z_i^{\star,H} - z_i^\star\|_1 \le \gamma^H$, the $\ell_1$ norm of the average follows the same inequality due to convexity, and we can bound the term using Hölder's inequality given $\|\theta_\star - c\|_\infty \le 2$ (see details of this result in Appendix F). To verify that the choice of $H$ in the theorem statement is appropriate, we can upper bound the 2nd term as

$$
2\gamma^H = 2((1-(1-\gamma))^{\frac{1}{1-\gamma}})^{\log(6/\epsilon)} \le 2e^{-\log(6/\epsilon)} = \tfrac{\epsilon}{3}.
$$

For the 3rd term, fixing $\theta_\star$ and $c$, the partial sum $\sum_{j=1}^i (\theta_\star - c)^\top(z_i^{\star,H} - \hat{z}_i^{\star,H})$ is a martingale. Since $\|z_i^{\star,H}\|_1 \le 1$, $\|\hat{z}_i^{\star,H}\|_1 \le 1$, and $\|\theta_\star - c\|_\infty \le 2$, we can initiate Lemma 3 by letting $b = 4$, and setting $n$ to sufficiently large to guarantee that the 3rd term is upper bounded by $\epsilon/3$ with high probability.

Given $(\theta_\star - c)^\top(\frac{\bar{Z}^\star}{n} - \frac{\bar{Z}}{n}) > \epsilon/3$, we can follow exactly the same analysis as for Theorem 2 to show that $B_\infty(\theta_\star, \epsilon/6)$ is never eliminated, and the number of updates can be bounded by $2d(d+1)\log\frac{12\sqrt{d}}{\epsilon}$. The number of total mistakes is the number of updates multiplied by $n$, the size of the mini-batches. Via Lemma 3, we can verify that the choice of $n$ in the theorem statement satisfies $|\sum_{j=1}^i (\theta_\star - c)^\top(z_i^{\star,H} - \hat{z}_i^{\star,H})| \le n\epsilon/3$ with probability at least $1 - \delta/\left(2d(d+1)\log\frac{12\sqrt{d}}{\epsilon}\right)$. Union bounding over all updates and the total failure probability can be bounded by $\delta$. $\qquad\square$