[Reviews · NeurIPS 2017]

Reviewer 1



The authors present a learning framework for inverse reinforcement learning wherein an agent provides policies for a variety of related tasks and a human determines whether or not the produced policies are acceptable or not. They present algorithms for learning a human's latent reward function over the tasks, and they give upper and lower bounds on the performance of the algorithms. They also address the setting where an agent's is "corrected" as it executes trajectories. This is a comprehensive theoretical treatment of a new conceptualization of IRL that I think is valuable. I have broad clarification/scoping questions and a few minor points. The main criterion for suboptimality given in (2) is relative to the initial state distribution, and a given \gamma. In Section 7, l294, suboptimality is given in terms of Q^\star (which is not explicitly defined but I take to mean the Q function for the policy of the human.) The authors state that "...whenever a suboptimal action is spotted in state s, it indeed implies that the agent's policy is suboptimal for s as the initial state. Hence, we can run Algorithm 1 and Theorem 2 immediately applies." What confuses me is it seems that this could happen for a state that has small (or zero) mass in the initial state distribution, and that might take a while to get to depending on P, and hence seeing this mistake along a trajectory may have little impact on (2) and may not indicate that the policy is suboptimal in that sense. Can the authors explain if they are being more stringent in this section, or what the connection is to the first formulation? I think it would help to have a concrete example -- and I am not asking for actual experiments here -- but a crisper example that the reader could latch onto as the different algorithms are discussed. The examples in the introduction from lines 18 through 28 are good motivation, but in my opinion they are too broad to help the reader really understand how these algorithms might work in practice. (In my opinion they are a little too "strong AI" to be helpful.) If the authors are going to argue in lines 143--149 about what requirements on human supervision are "reasonable," I think that would be better supported with a concrete example. Minor comments: l.4 - "surprised the agent" - surprised, the agent l.23 - goal but - goal, but l.38 - I know what you mean, but I don't know if \Delta(S) is standard enough to use w/o definition l.96 - I think "behaviourally equivalent in *all* MDP tasks" or similar might help clarify this statement for the reader l.301 - "using optimal policy" - "using the optimal policy" (or an optimal policy)

Reviewer 2



## Summary This is a theoretical work in which the authors present a new problem domain called repeated inverse reinforcement learning (RIRL). This domain describes interactions between a human (expert) and the RIRL agent, whereby the agent repeatedly presents solutions to tasks (in the form of a discounted state occupancy) and the human responds with a demonstration if surprised by the (suboptimality) of the agent's choice. At each iteration the agent also potentially makes some choice about the domain, which can include the state transition structure, and/or the task specific part of the reward function. The goal is to determine the task independent reward function, which in real terms might refer to such things as norms, preferences and safety considerations. Towards the end of the paper, the authors relax the requirement for a full state occupancy to be presented, instead allowing trajectories to be presented. The solution is presented either as an optimal policy for any desired task, or by identifying the task independent reward function up to an equivalence class. The main contributions of the paper are two algorithms and a series of convergence bounds proven for each setting. The algorithms making use of a 'volume reduction in ellipsoid' algorithm from the optimization literature, which reduces the space of possible reward functions to a minimal-volume enclosing ellipsoid. The paper is quite dense in theoretical results and pushes a number of proofs to appendices. Also, because of space considerations there are some very compact descriptions with high level intuition only. However, these results appear to be sound (to the best of my ability to judge) and to have general applicability in the newly defined problem domain. The authors could do a little more to provide both a concrete example to motivate the domain and some intuition to guide the reader (please see my comments below). Also, the tractability of this approach for 'interesting' domains is difficult to judge at this early theoretical stage. ## More detailed comments # p2 # Authors use \Delta to represent a probability distribution over various sets, but don't define this notation anywhere, or scope its meaning (some authors forbid continuous sets, other authors forbid discrete distributions with zero components when using this notation). # Or the identity function used for the state occupancy vector. # p4, line 158 Each task is denoted as a pair (X, R). # The authors do not describe the meaning of R in the general bandit description. From later text it becomes apparent, but it should be stated here explicitly. # p5, lines 171-177 # Is this really a generalisation? It seems more like a constraint to me. It certainly implies a constraint on the relationship between rewards for two different states which share features. Extension/modification might be a better word than generalisation. # It is also implied, but not stated that the policy now maps from features to actions, rather than from states to actions. # p5 line 183, language ...for normalization purpose, # for normalization purposes, # p5, line 186, meaning ...also contains the formal protocol of the process. # It is unclear what the authors mean by this. # p5 line 197, clarity Therefore, the update rule on Line 7... # on Line 7 of Algorithm 1 # p5 line 205, clarity ...in the worst case... # I think 'at it's smallest' (or similar) would be clearer. As this 'worst case' would represent the most accurate approximation to theta* possible after . A good thing. # p5 line 206, clarity To simplify calculation, we relax this l∞ ball to its inscribed l2 ball. # This took me a little while to interpret. It would be clearer if the authors talked about the lower bound represented by the l∞ ball being relaxed, rather than the ball being relaxed. # p6 line 208 and Equation below # When the authors say 'The unit sphere', I guess they are now talking about the l2 norm ball. This should be explicit. As should the vol function representing the volume of an ellipsoid. Also some intuition could be given as to why the third part of the inequality is as it is. If I am right then C_d sqrt(d)^d is the volume of an l2 ball that contains \Theta0 and C_d (ε/4)^d is the volume of the aforementioned lower bound l2 ball. # p7 meaning We argue that this is not a reasonable protocol for two reasons: (1) in expectation, the reward collected by the human may be less than that by the agent, which is due to us conditioning on the event that an error is spotted # Is there a way of making this statement clearer. # p7 clarity demonstration were still given in state occupancy # Is the meaning 'demonstration were still given in terms of a state occupancy vector' # and '(hence μ t )' could be expanded to '(we call the resulting state occupancy μ t ). # p8 discussion for Algorithm 2, intuition. # The authors could give some intuition behind the construction of the batch state vectors \bar{Z} and \bar{Z}^* in the Algorithm. The former appears to be an unnormalised state occupancy following \pi_t = \pi_{t-n} whose initial state distribution is uniformly sampled from the states with mistakes in between iterations t-n and t. Likewise, the \bar{Z}* vector is an unnormalised sum of discounted state visits from the n demonstrations initiated by the mistakes. Is this correct?

Reviewer 3



This is a theoretic paper that focuses on the repeated Inverse Reinforcement Learning problem proposed by the authors. The problem is interesting itself. The authors describe the problem and the challenge of identifying reward functions, and propose an identification algorithm for agent choose the tasks when it observes human behavior. The authors also provide an upper bound of the number tasks to reach desired accuracy when agent chooses the tasks, and also a lower bound on the number of mistakes that are inevitable in the worst case when nature chooses the tasks. The authors also propose a trajectory version of the ellipsoid algorithm for MDPs. The problem and theories described in this paper are novel and interesting. The paper is well written. It will be better there are some empirical results.